# ARPES detection of superconducting gap sign in unconventional superconductors

Qiang Gao[1,9], Jin Mo Bok[2,9], Ping Ai[1,9], Jing Liu[1,3,9], Hongtao Yan[1], Xiangyu Luo[1,4], Yongqing Cai[1], Cong Li [1], Yang Wang[1], Chaohui Yin[1,4], Hao Chen[1,4], Genda Gu[5], Fengfeng Zhang[6], Feng Yang[6], Shenjin Zhang[6], Qinjun Peng[6], Zhihai Zhu [1,4,7], Guodong Liu[1,4,7], Zuyan Xu[6], Tao Xiang [1,3,4], Lin Zhao [1,4,7]✉, Han-Yong Choi[8]✉ & X. J. Zhou [1,4,7]✉

The superconducting gap symmetry is crucial in understanding the underlying superconductivity mechanism. Angle-resolved photoemission spectroscopy (ARPES) has played a key role in determining the gap symmetry in unconventional superconductors. However, it has been considered so far that ARPES can only measure the magnitude of the superconducting gap but not its phase; the phase has to be detected by other phase-sensitive techniques. Here we propose a method to directly detect the superconducting gap sign by ARPES. This method is successfully validated in a cuprate superconductor $Bi_2Sr_2CaCu_2O_{8+\delta}$ with a well-known $d$-wave gap symmetry. When two bands have a strong interband interaction, the resulted electronic structures in the superconducting state are sensitive to the relative gap sign between the two bands. Our present work provides an approach to detect the gap sign and can be applied to various superconductors, particularly those with multiple orbitals like the iron-based superconductors.

The superconducting gap is the most basic and important physical quantity of superconductors which is characterized by the magnitude and sign. Its determination is essential for understanding the mechanism of superconductivity. While the conventional superconductors exhibit an $s$-wave gap symmetry that has the same sign along the entire Fermi surface, in unconventional superconductors, the superconducting gap may have different signs on different parts of the Fermi surface[1]. High temperature cuprate superconductors have been extensively studied for more than thirty years due to its unusually high critical temperature ($T_c$), anomalous normal state, and challenging mechanism of high temperature superconductivity[2–6]. One of the most significant achievements is the establishment of the $d$-wave

pairing symmetry that pinpoints the unconventional superconductivity mechanism in the cuprate superconductors[2]. For the magnitude of the superconducting gap, angle-resolved photoemission spectroscopy (ARPES) played an important role in directly determining the anisotropic gap size in the momentum space that is consistent with the $d$-wave symmetry[4,7,8]. However, the ARPES measurements do not provide the sign information that is necessary in pinning down the pairing symmetry. For the phase information of the $d$-wave gap, it was obtained later on by the phase-sensitive experiments based on Josephson tunneling and flux quantization[2,9,10] that were specially designed for the cuprate superconductors which have relatively simple Fermi surface and $d$-wave superconducting gap (Fig. 1a). These

[1]Beijing National Laboratory for Condensed Matter Physics, Institute of Physics, Chinese Academy of Sciences, Beijing 100190, China. [2]Department of Physics, Pohang University of Science and Technology (POSTECH), Pohang 37673, Korea. [3]Beijing Academy of Quantum Information Sciences, Beijing 100193, China. [4]School of Physical Sciences, University of Chinese Academy of Sciences, Beijing 100049, China. [5]Condensed Matter Physics and Materials Science Department, Brookhaven National Laboratory, Upton, NY 11973, USA. [6]Technical Institute of Physics and Chemistry, Chinese Academy of Sciences, Beijing 100190, China. [7]Songshan Lake Materials Laboratory, Dongguan, Guangdong 523808, China. [8]Department of Physics and Institute for Basic Science Research, SungKyunKwan University, Suwon 440-746, Korea. [9]These authors contributed equally: Qiang Gao, Jin Mo Bok, Ping Ai, Jing Liu. ✉e-mail: LZhao@iphy.ac.cn; hychoi@skku.edu; XJZhou@iphy.ac.cn

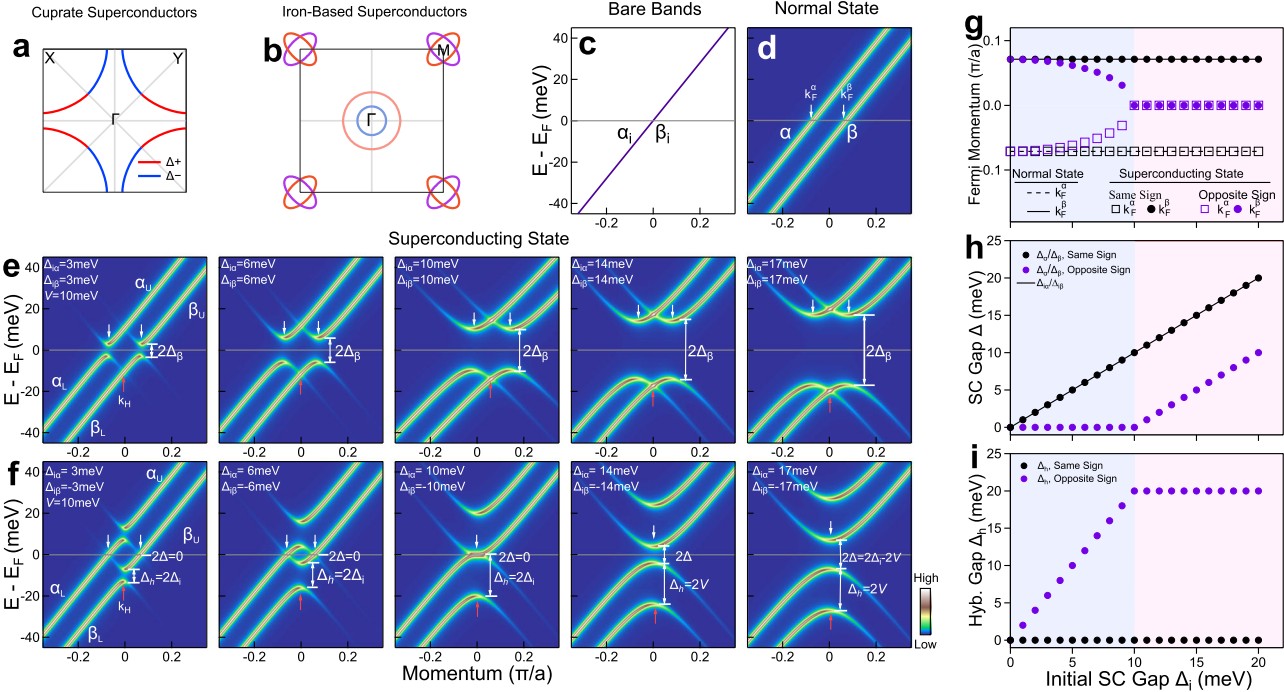

**Fig. 1 | The proposal to detect the relative superconducting gap sign between two bands. a** Schematic Fermi surface of cuprate superconductors. The superconducting gap has a $d$-wave form which exhibits a sign change along the nodal directions. **b** Typical Fermi surface of the iron-based superconductors with multiple hole and electron pockets. **c–i** Proposed method to determine the relative sign of the superconducting gaps between two bands. **c** shows the initial two bare bands that are degenerate ($\alpha_i$ and $\beta_i$). **d** shows the simulated $\alpha$ and $\beta$ bands in the normal state after putting the interband coupling ($V = 10$ meV) with the Fermi momenta marked. **e** shows the simulated band structures in the superconducting state with a fixed $V = 10$ meV and increasing initial gap size $\Delta_i$. Here, the same magnitude and the same sign of the initial superconducting gaps are taken for the two bands. Superconducting gap ($2\Delta_\alpha$ and $2\Delta_\beta$) opens at the Fermi momenta $k_F^\alpha$ and $k_F^\beta$, yielding four branches of bands labeled as $\alpha_U$, $\beta_U$, $\alpha_L$ and $\beta_L$. The Bogoliubov backbending band

of $\alpha_L$ crosses the $\beta_L$ band at a momentum $k_H$ as marked by red arrows. **f** Same as **e** but the opposite sign of the initial superconducting gaps are taken for the two bands. The Fermi momentum is no longer conserved from the normal state. When defined as the momentum that is close to the Fermi level, the Fermi momentum changes dramatically with the variation of $\Delta_i$. The superconducting gap becomes also significantly different from the general picture in **e**. Pronounced Bogoliubov hybridization gap ($\Delta_h$) opens at $k_H$ in this case while it is basically zero in **e**. **g** Evolution of the Fermi momenta of the $\alpha$ and $\beta$ bands with $\Delta_i$ in the normal and superconducting states where the same and opposite signs of the two initial superconducting gaps are considered. **h** Evolution of the superconducting gap $\Delta_\alpha$ and $\Delta_\beta$ with $\Delta_i$ when the two initial gaps have the same sign and opposite sign. **i** Evolution of the hybridization gap $\Delta_h$ with $\Delta_i$ when the two initial gaps have the same sign and opposite sign.

methods have not been successfully applied to the other superconductors like the iron-based superconductors which possess multiple Fermi surface sheets and possible unconventional pairing with gap sign changes (Fig. 1b)[11–13]. So far, experimental extraction of the sign information in the gap function has proven to be significant but challenging in unconventional superconductors[1]. It has been attempted in the scanning tunneling microscopy (STM) measurements utilizing quasiparticle interference[14–18]. Although ARPES is a powerful tool to directly measure the superconducting gap magnitude, it has long been believed that it can not probe the sign of the superconducting gap and therefore there has been no ARPES report in the sign detection of the superconducting gap.

In this work, we develop a method to detect superconducting gap sign by ARPES. It is motivated by our observation of an unusual Bogoliubov band hybridization in the cuprate superconductor Bi$_2$Sr$_2$CaCu$_2$O$_{8+\delta}$ (Bi2212)[19]. The gap sign manifests itself in the resulted electronic structures in the superconducting state including the Fermi momentum shift, the strong Bogoliubov band hybridization and the abnormal superconducting gap behaviors. The proposed method is well tested in the ARPES measurements of Bi2212 with a $d$-wave gap symmetry. The present work provides the approach to detect the superconducting gap sign which is significant for understanding the mechanism of unconventional superconductors.

## Results
### Proposed method
To detect the relative sign of the superconducting gap between two bands, we propose a method based on the Bogoliubov band hybridization. For a system with two bands ($\alpha$ and $\beta$) which are close in momentum space (Fig. 1c–f), its superconducting state can be described by a phenomenological Hamiltonian

$$\hat{H}(\mathbf{k}) = \begin{pmatrix} \varepsilon_{i\alpha} & V & \Delta_{i\alpha} & 0 \\ V & \varepsilon_{i\beta} & 0 & \Delta_{i\beta} \\ \Delta_{i\alpha} & 0 & -\varepsilon_{i\alpha} & -V \\ 0 & \Delta_{i\beta} & -V & -\varepsilon_{i\beta} \end{pmatrix} \quad (1)$$

where $\varepsilon_{i\alpha}$ and $\varepsilon_{i\beta}$ represent the initial $\alpha$ and $\beta$ bare bands, $V$ is the coupling strength between the two bands, and $\Delta_{i\alpha}$ and $\Delta_{i\beta}$ are the initial superconducting gap of the $\alpha$ and $\beta$ bands[20,21]. Such a Hamiltonian can also describe the normal state when the initial superconducting gaps are taken as zeros.

Figure 1c–i shows the simulated band structures of the two band system in the normal state and in the superconducting state (see Supplementary Note 1 for the details of the simulation). To be typical and for simplicity, we started with the two initial bare bands which are

degenerate (Fig. 1c). When there is an interband coupling ($V = 10$ meV) between the two bands, the band structure in the normal state exhibits a band splitting (Fig. 1d). In the superconducting state, the relative sign of the initial superconducting gap between the two bands dramatically affects the resulted band structures. When the initial superconducting gap of the two bands is taken as the same both in the magnitude and in its sign (Fig. 1e), superconducting gap opens in the normal way at the two Fermi momenta and there is no noticeable hybridization between the Bogoliubov backbending band of $\alpha$ and the $\beta$ band at the crossing point $k_H$. In contrast, when the initial superconducting gap of the two bands takes the opposite sign, the resulted band structures (Fig. 1f) become totally different from those in Fig. 1e in terms of the unusual change of the Fermi momentum (Fig. 1g), the superconducting gap (Fig. 1h) and the opening of a hybridization gap at $k_H$ (Fig. 1i). While the two Fermi momenta keep fixed in the superconducting state when the two initial superconducting gaps have the same sign, they can be dramatically shifted in the case of the opposite gap sign (Fig. 1g). Depending on the relative magnitude of the initial superconducting gap ($\Delta_i$) and the interband coupling ($V$), the two Fermi momenta in the normal state may even evolve into the same one in the superconducting state. In the case of the superconducting gap, although it keeps the same with the initial superconducting gap in the superconducting state when the two initial superconducting gaps have the same sign, it is completely altered when the two initial superconducting gaps take the opposite sign (Fig. 1h). The superconducting gap may even become zero in the superconducting state when the initial superconducting gap ($\Delta_i$) is relatively smaller than the interband coupling ($V$). Whereas the band hybridization gap is nearly zero when the two initial superconducting gaps have the same sign, it can become significant in the case of the opposite gap sign in the superconducting state (Fig. 1i). These fundamental differences in the resulted band structures form the basis of our proposed method to probe the relative gap sign between the two bands through the ARPES measurements.

We just discussed one typical and extreme case where the initial two bare bands are degenerate. We also carried out simulations on the cases that the initial two bare bands are different, in particular, how the Bogoliubov band hybridization evolves with the separation of the two bare bands (Supplementary Fig. 1). It is found that the dramatic difference of the electronic structures in the superconducting state between the same-gap-sign and the opposite-gap-sign cases still persists. Even if the initial two bare bands separate, when the initial two superconducting gaps have the opposite sign, the unusual behaviors like the Fermi momentum shift, the pronounced Bogoliubov band hybridization and the abnormal superconducting gap remain present. This indicates that our proposed method is more general which can be used for both cases that the initial two bare bands are degenerate and separate.

### Manifestations of gap sign in Bogoliubov band hybridization

In order to test the above proposed method, it is necessary to find a superconductor that is unconventional and its superconducting gap symmetry is well established with a sign change. Such superconductors are rare and cuprate superconductors are essentially the only known case that can satisfy the stringent requirements. It is well known that the cuprate superconductors have a $d$-wave superconducting gap that changes sign on different parts of the Fermi surface (Fig. 1a). Furthermore, to test the method in the cuprate superconductors, it is also necessary to find two bands that are close in momentum space and have the opposite gap sign.

In the process of studying the origin of the superstructure bands in Bi2212[19], we came across an ideal case that can test our proposed method. In Bi2212, because of the presence of the incommensurate structural modulations, in addition to the main bonding and antibonding bands, superstructure bands are formed by shifting the main bands with the superstructure wavevector, $\pm \mathbf{Q}$, as shown in Fig. 2a and

Supplementary Fig. 2[19,22,23]. While the main Fermi surface and the superstructure Fermi surface are well separated in the first quadrant, they cross each other in the second quadrant (Fig. 2a). Figure 2b shows a constant energy contour near the Fermi level covering the band crossing area in the second quadrant. Here mainly two Fermi surface sheets are observed due to the photoemission matrix element effects: the main antibonding Fermi surface sheet(AB) and the superstructure antibonding Fermi surface sheet(AB_SS). It has been found that the main bands and the superstructure bands exhibit a selective band hybridization, i.e., the initial main antibonding band (AB, red line in Fig. 2b) hybridizes with the initial superstructure bonding band (BB_SS, dashed blue line in Fig. 2b)[19,24], as shown in Fig. 2b. The main AB Fermi surface is then broken into two branches (Branch1 and Branch2) at the crossing point MS induced by the hybridization. Therefore, we have found a rare but ideal case to test our method in Bi2212: (1), There are two bands, the main antibonding band (AB) and the superstructure bonding band (BB_SS), that are close-by in momentum space; (2), These two bands exhibit strong interband coupling; (3), The superconducting gap sign of the two bands is opposite; and (4), The superconducting gap magnitude of the two bands is similar.

Figure 2 shows the band structures of Bi2212 measured in the normal state (Fig. 2c) and the superconducting state (Fig. 2d) along a typical momentum cut near the crossing region of the AB and BB_SS Fermi surface. In the normal state, two bands are mainly observed, labeled as BR1 and BR2 in Fig. 2c, that correspond to Branch1 and Branch2 Fermi surface in Fig. 2b. In the superconducting state, the observed band structure (Fig. 2d) is strikingly different from generally expected picture that superconducting gaps open at the two Fermi momenta. It is unusual in several aspects. First, there is an obvious Fermi momentum shift in the superconducting state. The Fermi momentum separation between the two bands shrinks from $0.023\pi/a$ in the normal state to $0.018\pi/a$ in the superconducting state. Second, the gap opening at the two Fermi momenta is quite unusual. The particle-hole symmetry is not conserved at the Fermi momentum as seen from the BR2 band in Fig. 2d and the photoemission spectrum (energy distribution curve, EDC) at the BR2 Fermi momentum (purple line in Fig. 2e). The gap center is obviously away from the chemical potential. This is not caused by the improper Fermi function division[25] because the energy resolution we used is rather high (~1 meV). Third, below the Fermi level, the BR2 band breaks into two parts with a strong spectral weight suppression around the binding energy of 14 meV, as marked by red arrows in Fig. 2d. Such a band breaking and the dramatic spectral weight suppression of the BR2 band can also be seen from the corresponding EDCs in the superconducting state (Fig. 2e).

The unusual behaviors observed in the superconducting state (Fig. 2d) can be understood in terms of the two band model we proposed in Equation (1) by taking proper bare bands, initial superconducting gaps and the coupling strength between the two bands. Figure 2f–i shows the simulated band structure in the normal state and superconducting state. In the simulation process, the relative sign of the initial superconducting gap between the two bands plays a decisive role in dictating the band structures in the superconducting state. Figure 2h shows the simulated band structure in the superconducting state by considering the opposite sign of the superconducting gaps on the two bands. The corresponding EDCs are shown in Fig. 2j. The simulated results (Fig. 2h, j) are highly consistent, even on the quantitative level, with the measured band structure (Fig. 2d) and EDCs (Fig. 2e). All the unusual behaviors observed in the superconducting state are well captured by the simulations. In contrast, if the same sign of the superconducting gap is taken for the two bands, the simulated band structure (Fig. 2i) deviates far from the measured result (Fig. 2d). The observation of the unusual band structures in the superconducting state and their quantitative understanding based on the two band model indicate unambiguously that the superconducting gaps of the two bands have the opposite sign. It demonstrates the

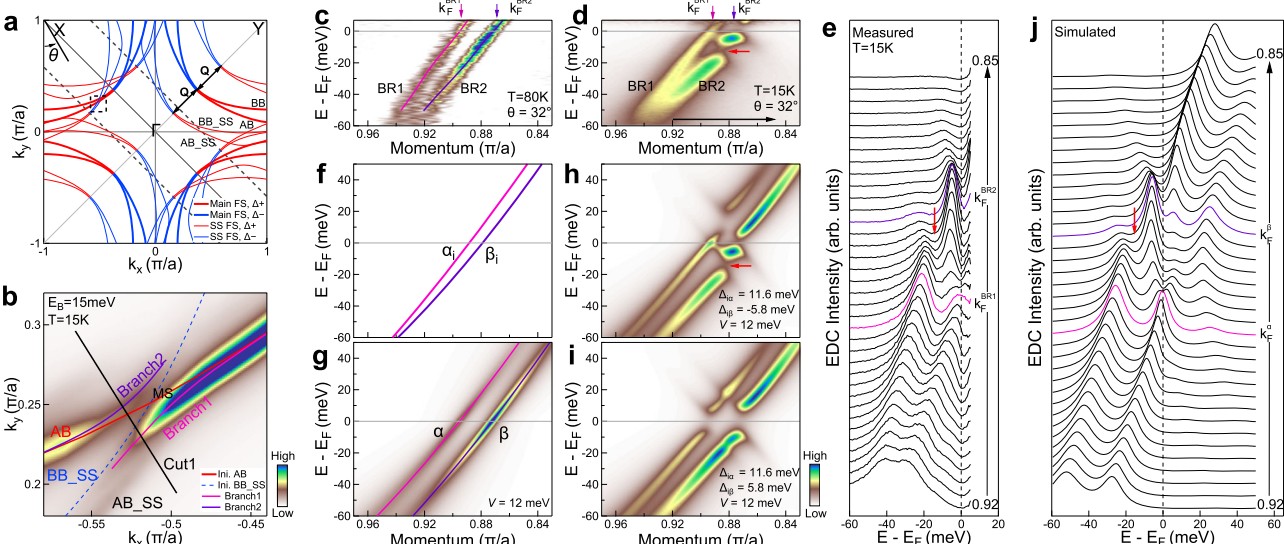

**Fig. 2 | Signatures of the opposite gap sign on the two bands in Bi2212.**
**a** Schematic Fermi surface of Bi2212 that is composed of the main Fermi surface (thick lines, antibonding (AB) and bonding (BB)) and the superstructure Fermi surface (thin lines, AB_SS and BB_SS). The superstructure Fermi surface is obtained by shifting the main Fermi surface with a superstructure wave vector ± **Q** along Γ-Y. The superconducting gap sign is marked by blue (positive) and red (negative). **b** Constant energy contour at the binding energy of 15 meV measured at 15 K. The covered momentum space (dashed square in **a**) lies in the second quadrant. The initial AB (Ini. AB) and BB_SS (Ini. BB_SS) cross at a point MS. The main AB is broken into two branches (Branch1 and Branch2) at MS due to its hybridization with the superstructure BB_SS. **c** Band structure in the normal state measured at 80 K along the momentum cut Cut1. It is a second derivative image with respect to the momentum. The momentum cut position is marked in **b** by the black line with a

Fermi surface angle $\theta$ = 32, as defined in **a**. Two bands are observed corresponding to the Branch1 (BR1, pink line) and Branch2 (BR2, purple line) Fermi surface in **b**. The origin of the momentum axis is defined at the zone corner. We adopt this definition throughout the paper. **d** Same as **c** but measured in the superconducting state at 15 K. The image is acquired by dividing the Fermi distribution function. **e** The energy distribution curves (EDCs) of **d**. **f–i** Simulated band structures with an interband coupling. **f** shows two initial bare bands without band coupling. **g** shows two bands in the normal state after putting $V$ = 12 meV. **h** shows the band structure in the superconducting state. The initial superconducting gaps of the two bands are assumed to be $\Delta_{i\alpha}$ = 11.6 meV and $\Delta_{i\beta}$ = −5.8 meV with the opposite sign. **i** Same as **h** but the initial superconducting gaps are assumed to have the same sign: $\Delta_{i\alpha}$ = 11.6 meV and $\Delta_{i\beta}$ = 5.8 meV. **j** EDCs of **h**.

feasibility of our proposed method in detecting the relative sign of the superconducting gaps on two bands. This is the first time that the superconducting gap sign is detected from ARPES measurements.

The relative gap sign also manifests itself in the Fermi surface topology, momentum dependence of the band structure, and the associated momentum dependence of the Bogoliubov band hybridization. Figure 3a shows the detailed Fermi surface mapping in the superconducting state around the crossing point MS of the initial main AB and the superstructure BB_SS Fermi surface. The hybridization between the AB and BB_SS bands results in breaking the main AB Fermi surface into two branches (Branch1 and Branch2) with the spectral weight enhanced near the crossing point. Figure 3b shows band structures measured along different cuts in the covered momentum space in Fig. 3a, c (more complete momentum-dependent band structures are shown in Supplementary Fig. 3). Within a narrow momentum space, the observed band structures exhibit a dramatic and systematic momentum dependence. Moreover, like the band structure for $\theta$ = 32 that is analyzed in detail in Fig. 2, these observed bands are also unusual in the gap opening and Bogoliubov band hybridization. The BR1 band shows a strong Bogoliubov hybridization with the BR2 band near the crossing point MS (Cut1 and Cut2 in Fig. 3b) and the hybridization gets weaker with the momentum cuts moving away from MS (Cut3 to Cut5 in Fig. 3b). The quantitative evolution of the Bogoliubov hybridization gap with momentum is extracted in Supplementary Fig. 4.

The measured Fermi surface mapping (Fig. 3a) and the unusual momentum-dependent band structure evolution (Fig. 3b) can be understood by the two band model (Equation (1)) only when the relative sign of the superconducting gaps on the two bands are taken opposite. We note that, in the particular case of Bi2212, the initial two bare bands and the initial two superconducting gap sizes are known

beforehand which can be determined from the measurements on the two main bands (AB and BB) in the first quadrant (see Supplementary Fig. 5, Supplementary Fig. 6 and the tight binding fitting in Supplementary Note 2). In principle, the coupling strength $V$ between the main AB band and the superstructure BB_SS band can also be determined from the band structure measurements in the normal state[19]. Here we take it as a constant in the small covered momentum space of the Fermi surface crossing region. Under the condition that all the parameters in Equation (1) are known, we globally simulated the Fermi surface and momentum-dependent band structures by considering the opposite gap sign (Fig. 3d, e) and the same gap sign (Fig. 3f, g) for the two bands. When the two gaps take the opposite sign, the simulated Fermi surface mapping (Fig. 3d) well reproduces the measured Fermi surface in Fig. 3a. The momentum-dependent band structures and the associated Bogoliubov band hybridization (Fig. 3b) are well captured in the simulated results (Fig. 3e). In particular, the measured momentum-dependent Bogoliubov hybridization gap (black circles in Fig. 3h) shows a quantitative agreement with the simulations (red line in Fig. 3h). In contrast, if the same gap sign is taken for the two bands, the simulated Fermi surface (Fig. 3f), the overall momentum-dependent band structures (Fig. 3g) and the Bogoliubov hybridization gap (blue line in Fig. 3h) deviate significantly from the measured results. These results lend further decisive evidence that the relative sign of the superconducting gap on the two bands is opposite in the covered momentum space.

**Manifestations of gap sign in the abnormal superconducting gap**
As shown from the simulations of the two band model in Fig. 1 and Supplementary Fig. 1, one of the main signatures of the relative gap sign is the unusual superconducting gap behaviors. Such an anomaly of the superconducting gap is observed in Bi2212 in the crossing area

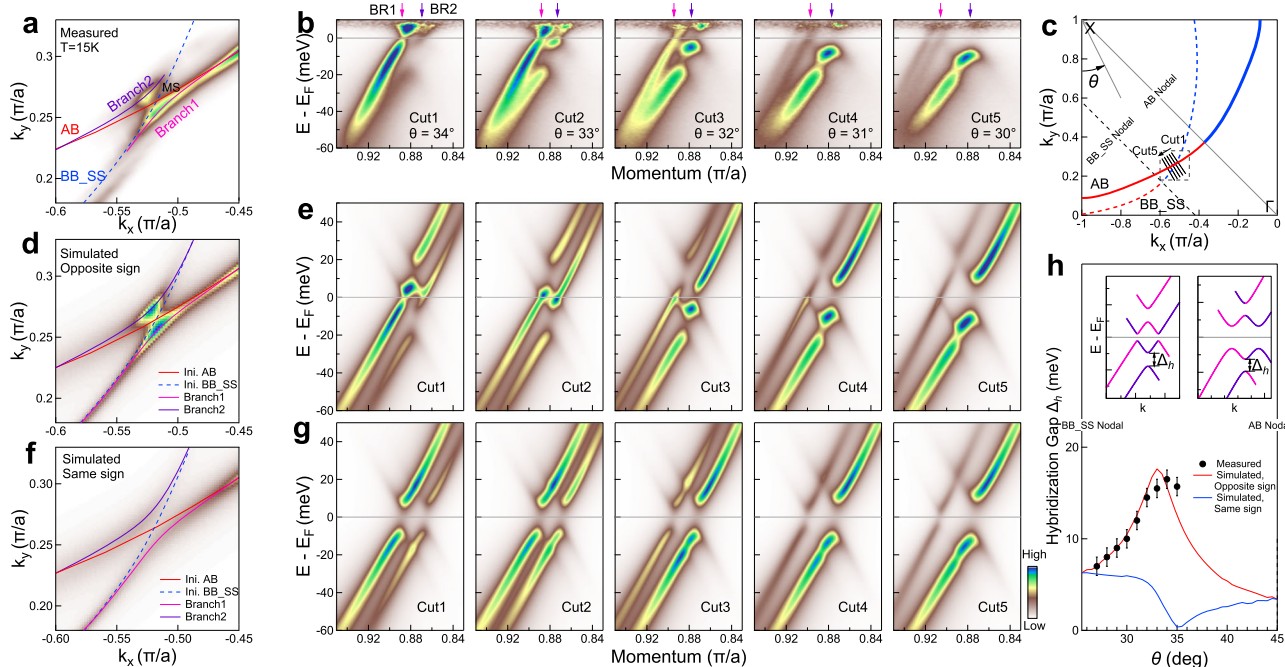

**Fig. 3 | Signatures of the opposite gap sign in the momentum dependence of the Bogoliubov band hybridization in Bi2212. a** Fermi surface mapping of Bi2212 measured at 15 K. The covered momentum space is in the second quadrant as shown by the dashed black frame in **c**. The AB Fermi surface is mainly observed but is broken into two branches, Branch1 (pink line) and Branch2 (purple line), due to hybridization with the superstructure BB_SS Fermi surface. **b** Band structures measured along different momentum cuts around the crossing area of the initial Fermi surface AB and BB_SS. The images are obtained by dividing the Fermi distribution function to show part of the band structure above the Fermi level. The location of the momentum cuts is shown by the solid black lines in **c** and also be defined by the Fermi surface angle, $\theta$. Two bands are mainly observed as marked by pink arrows for the Branch1 (BR1) and purple arrows for the Branch2 (BR2). **c** Schematic Fermi surface in the second quadrant with the related initial main AB

Fermi surface and the superstructure BB_SS Fermi surface plotted. The gap sign on the Fermi surface is marked by the red color for the positive and the blue color for the negative by considering $d$-wave pairing symmetry. **d** Simulated Fermi surface of Bi2212 by considering the opposite gap sign on the AB and BB_SS Fermi surface in the Fermi surface crossing area. **e** The corresponding simulated band structures along different momentum cuts. **f**, **g** Same as **d**, **e** but by considering the same gap sign on the AB and BB_SS Fermi surface. **h** Momentum dependence of the Bogoliubov hybridization gap $\Delta_h$. The definition of $\Delta_h$ is schematically shown in the upper insets. The measured values (see Supplementary Fig. 4) are plotted as black circles. The red (blue) line shows the simulated $\Delta_h$ by considering the opposite (same) gap sign of the main band AB and the superstructure band BB_SS. Error bars are determined based on the fitting error of the EDC peak position.

of two bands with opposite gap sign (Fig. 4a). Figure 4b highlights the measured band structure near the Fermi level along the momentum cut of $\theta$=32 in the superconducting state. The corresponding original EDCs and the EDCs after dividing the Fermi distribution function at the two Fermi momenta (BR1_$k_F$ and BR2_$k_F$) are shown in Fig. 4c–f, respectively. The superconducting gaps of the two bands are unusual in several aspects. First, the BR1 band crosses the Fermi level with a gap size that is nearly zero. Second, the particle-hole symmetry for the BR2 band is apparently broken where the spectral weight at the Fermi momentum is not symmetrical with respect to the Fermi level. Third, compared with the gap size at the equivalent momentum position in the first quadrant, the gap size of the BR1 and BR2 bands after the hybridization in the second quadrant is much reduced from the initial 5.8 meV and 11.6 meV to 0 meV and 5 meV, respectively. As shown in Fig. 2h, these unusual gap behaviors can be well reproduced when the relative gap sign of the two bands is opposite. We note that the particle-hole symmetry breaking manifests itself in the dramatic difference of the spectral weight distribution with respect to the chemical potential but the energy positions remain symmetric with respect to the chemical potential. Since the superconducting gap size is determined from the EDC peak position, the broken particle-hole symmetry does not affect our determination of the superconducting gap even near the nodal region.

Figure 4g, h shows EDCs measured along the BR1 and BR2 Fermi surface in the superconducting state. The momentum dependent superconducting gap along the two branches of Fermi surface is plotted in Fig. 4i. Near the crossing area, the measured

superconducting gap strongly deviates from the standard $d$-wave form with the gap size much reduced and even new accidental nodes can be produced. The unusual momentum dependence of the superconducting gap can be well understood by considering the two bands hybridization and the opposite gap sign on the two bands. As shown in Fig. 4i, the superconducting gap obtained from the same global band structure simulations (Fig. 3e) is quantitatively consistent with the measured results. This further demonstrates that the relative sign of the superconducting gap on the two bands is opposite in the covered momentum space.

## Discussion

Our present results show that the hybridization between two bands with opposite superconducting gap sign can induce unusual behaviors. First, the Fermi momenta can be shifted in the superconducting state relative to those in the normal state (Fig. 2c, d). Second, the density of states at the Fermi momenta may become unsymmetrical with respect to the Fermi level which indicates the particle-hole symmetry is no longer conserved in the superconducting state (Fig. 2e). Third, it can produce gap nodes even though the initial gap of the two bands is non-zero (Fig. 4e). Our results indicate that, because of the superstructures and band hybridization in Bi2212, additional accidental gap nodes or even segments of gapless Fermi surface can be produced besides the usual nodes along the nodal directions in the first Brillouin zone. These should be considered in detecting the gap symmetry of cuprate superconductors[18,26,27]. Our results also point to the possibility that some unusual superconductors may be designed

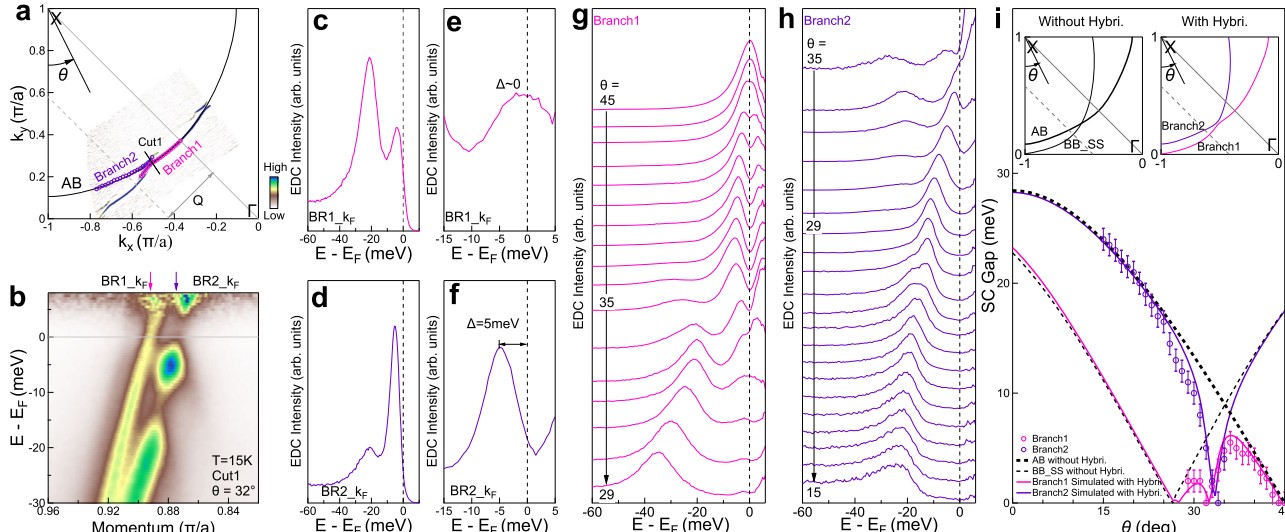

**Fig. 4 | Manifestation of the opposite gap sign in the unusual momentum-dependent superconducting gap of Bi2212. a** Fermi surface mapping of Bi2212 in the second quadrant measured at 15 K. It is the MDC second derivative image. **b** Band structure measured at 15 K along the momentum cut (Cut1 in **a**) with the Fermi surface angle $\theta = 32$. The image is obtained by dividing the Fermi distribution function. **c, d** Original EDCs at the Fermi momenta of the BR1 and BR2 bands, respectively, obtained from **b**. **e, f** Corresponding EDCs of **c, d** after dividing the Fermi distribution function. The superconducting gap size is nearly zero for the BR1 band and 5 meV for the BR2 band. **g, h** EDCs from the Branch1 and Branch2 parts of Fermi surface, respectively, obtained by dividing the Fermi distribution function. The location of the Fermi momentum points is shown by the pink open circles for the Branch1 and purple open circles for the Branch2 in **a** which is also defined by the angle $\theta$. **i** Momentum dependence of the superconducting gap on the two branches of Fermi surface. The superconducting gaps on the Branch1 (pink open circles) and

the Branch2 (purple open circles) are obtained from the EDCs in **g, h**. Error bars are determined based on the fitting error of the EDC peak position. The dashed thick black line represents the usual $d$-wave superconducting gap ($\Delta = 29 * |\cos k_x - \cos k_y|/2)$ on the AB Fermi surface as obtained in the first quadrant (Supplementary Fig. 6d). The dashed thin black line represents the usual superconducting gap on the BB_SS Fermi surface. Since the superstructure Fermi surface is shifted by a wavevector **Q** along the Γ-Y direction relative to the main Fermi surface, the nodal point in the second quadrant is shifted from the initial $\theta = 45$ for the main Fermi surface to $\theta = 26.5$ for the superstructure Fermi surface, as shown by the dashed thin black line in the top insets. The solid pink line and solid purple line represent the simulated superconducting gaps on the Branch1 and Branch2 Fermi surface, respectively, by considering the Bogoliubov band hybridization between the two bands.

---

and produced. As an extreme case, we consider a material that consists of two Fermi surface sheets originating from the initial two degenerate bands due to the interband interaction. As shown in Supplementary Fig. 7, if the superconducting gap sign is opposite on the two bands, it is possible that the superconducting gap becomes zero on both Fermi surface sheets even though the initial gap is non-zero. This would produce a gapless superconductor with zero superconducting gap on the entire Fermi surface. It is interesting to explore whether such unusual superconductors can be realized both theoretically and experimentally.

In unconventional superconductors, the determination of the gap symmetry, particularly the phase, is both significant and challenging[1,12]. In the cuprate superconductors, the phase information of the $d$-wave gap is obtained by the phase-sensitive experiments based on Josephson tunneling and flux quantization[2,9,10]. However, these methods were specifically designed for the cuprate superconductors which have relatively simple Fermi surface and $d$-wave superconducting gap (Fig. 1a). For the iron-based superconductors that possess multiple Fermi surface sheets and possible unconventional pairing with gap sign changes (Fig. 1b)[11–13], the conventional phase-sensitive methods based on Josephson tunneling and flux quantization have not been successful in determining the phase of the superconducting gap. The STM measurements have been tried to extract the phase information of the superconducting gap by utilizing quasiparticle interference but this method is limited to only a few materials[14–18]. In the present work, we proposed a method to detect the phase of the superconducting gap. It provides direct information of the relative gap sign on the two bands that are studied, regardless of the complexity of the multiple orbital systems. It is a general method that can be applied in various superconductors, including cuprate superconductors, iron-based

superconductors, heavy Fermion superconductors and so on. Since ARPES is good at directly determining the anisotropic gap size in the momentum space, with its expanded capability to provide the phase information, it will become a more powerful tool to determine the pairing symmetry of superconductors.

As shown in Fig. 1 and Supplementary Fig. 1, the applicability of our method depends on the significant difference of the band structures in the superconducting state when the initial superconducting gaps of the two bands have the same or opposite signs. In particular, when the initial superconducting gaps have the opposite sign, some distinct and unusual signatures may appear such as the Fermi momentum shift, the strong Bogoliubov band hybridization and the anomalous gap behaviors. In addition to the relative gap sign, the band structure in the superconducting state is determined by the relative position of the two bare bands, the interband coupling strength and the initial superconducting gap size. The most ideal case to use the method is that the two bare bands are degenerate or adjacent, the two initial gap sizes are the same or very close and the interband coupling is relatively strong. The momentum cut of $\theta = 33$ is close to such an ideal case (the second panel in Fig. 3b). In less ideal cases when the two bare bands are separated or the two initial gaps are different, the signatures of the relative gap sign become less prominent and can not be identified qualitatively. Careful quantitative analyses are needed to distinguish the relative gap sign. The momentum cuts of $\theta = 34$ and 32 belong to such less ideal cases (the first and third panels in Fig. 3b). In the cases when the two bare bands are far apart, or the two initial superconducting gaps are significantly different or the interband coupling strength is rather small, the difference induced by the relative gap sign becomes small in the resulted band structures and the method becomes ineffective. Since the effectiveness of the method

requires that the two bands are close and have relatively strong interband coupling, it is good for testing the proposed gap structure but is difficult to determine the full momentum-dependence of the phase. The utilization of the method requires an overall understanding of the band structure and superconducting gap structure of the involved two bands.

The Bogoliubov band hybridization between the two bands in Bi2212 or between the two bands from the inner and outer $CuO_2$ layers in $Bi_2Sr_2Ca_2Cu_3O_{10}$ (Bi2223)[20,21] can be universally described by the same two-band model in Equation (1). The extent of the hybridization depends on the separation of the two bands, their interband coupling strength, the superconducting gap size difference on the two bands, and their relative gap sign. In Bi2223, the observed strong Bogoliubov band hybridization between the bands from the inner and outer $CuO_2$ layers can be well understood by the large superconducting gap size difference of the two bands even though the gaps have the same sign[20,21]. In Bi2212, the main bonding band and the main antibonding band have comparable superconducting gap size[28] and the same gap sign as shown in the first quadrant of the Brillouin zone in Fig. 2a. According to the two-band model, the resulted Bogoliubov band hybridization should be nearly zero. This is consistent with the measured results that no obvious hybridization is observed between the two bands (Supplementary Fig. 5). On the other hand, the absence of the Bogoliubov band hybridization between the main bonding band and the main antibonding band indicates that they have the same gap sign. In Bi2212, the main antibonding Fermi surface and the superstructure bonding Fermi surface cross each other in the second quadrant of the Brillouin zone, as shown in Fig. 2a. Near the crossing region, the main antibonding band and the superstructure bonding band have identical gap size but the opposite gap sign. This gives rise to a dramatic Bogoliubov band hybridization between the two bands as shown in Figs. 2 and 3. In order to see the obvious Bogoliubov band hybridization effect, the local superconducting gaps should be different on the two bands. In the present case, the gap sign is opposite between the main antibonding band and the superstructure bonding band near the crossing region in the second quadrant.

Our proposed method works well when the two bands are close in the momentum space with a strong interaction between them. It can be used to test whether the two hole-like pockets around the zone center in the iron-based superconductors (Fig. 1b) have the same or opposite gap sign[29,30]. It can also be used to check whether the two electron-like pockets around M (Fig. 1b) have the same or opposite gap sign, particularly in the iron-based superconductors with only electron pockets[31–35]. When the two bands are far away in the momentum space, it is possible to use the method by taking advantage of the band folding from the superstructure or surface reconstruction. In the present work, although the gap sign change occurs on different parts of the Fermi surface, it is detected from the interaction of the main band and the folded band from the superstructures in Bi2212. In the iron-based superconductors like $(Ba,K)Fe_2As_2$, in order to determine the relative gap sign between $\Gamma$ and M, i.e., whether the gap symmetry is $s_\pm$ or $s_{++}$[36–41], the $\sqrt{2} \times \sqrt{2}$ surface reconstruction can be used to fold the bands between the $\Gamma$ and M points[42,43]. In the case that no natural band folding mechanisms are available, it is possible to engineer the band structures to produce hybridization between the initial bands and the superstructure bands such as those realized in twisted graphene[44–46] or substrate-controlled film growth[47,48]. It should be noted that the key requirement is a perturbative periodic potential that folds the native Fermi surface and thus creates crossing points. This is not far from practical in the age of van-der-Waals heterostructures[49].

In summary, we have proposed a method to detect superconducting gap sign by ARPES. The method is well tested in the ARPES measurements of Bi2212 with a known $d$-wave gap symmetry. The gap sign manifests itself in the resulted electronic structures in the superconducting state including the Fermi momentum shift, the Bogoliubov band hybridization and the abnormal superconducting gap behaviors. We have demonstrated that, in addition to measuring the superconducting gap size, ARPES can be phase sensitive and deterministic in pinning down the pairing symmetry that is significant for understanding the superconductivity mechanism of unconventional superconductors.

## Methods

ARPES measurements were carried out on a vacuum ultraviolet (VUV) laser-based ARPES system equipped with an electron energy analyzer (Scienta Omicron DA30L)[19,50,51]. The photon energy of the VUV laser is 6.994 eV. The overall energy resolution was set at 1.0 meV. The angular resolution was ~ 0.3°, corresponding to a momentum resolution of 0.004 $Å^{-1}$ with the photon energy of 6.994 eV. The Fermi level is referenced by measuring on the Fermi edge of a clean polycrystalline gold that is electrically connected to the sample. High quality overdoped $Bi_2Sr_2CaCu_2O_{8+\delta}$ ($T_c = 78$ K) single crystals were grown by the floating zone method and then annealed in oxygen atmosphere[52]. The $T_c$ was measured using a Quantum Design SQUID magnetometer. The sample was cleaved in situ and measured in vacuum with a base pressure better than $3 \times 10^{-11}$ mbar.

## Data availability

All data are processed by Igor Pro 8.0.2 software. All data needed to evaluate the conclusions in the paper are available within the article and its Supplementary Information files. All data generated during the current study are available from the corresponding author upon request.

## Code availability

The codes used for the calculations in this study are available from the corresponding authors upon request.

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

## Acknowledgements

This work is supported by the National Key Research and Development Program of China (No. 2021YFA1401800, 2017YFA0302900, 2018YFA0305600, 2018YFA0704200, 2019YFA0308000, 2022YFA1604200 and 2022YFA1403900), the National Natural Science Foundation of China (Grant No. 11888101, 11922414, 11974404 and 12074411), the Strategic Priority Research Program (B) of the Chinese Academy of Sciences (XDB25000000 and XDB33000000), the Innovation Program for Quantum Science and Technology (Grant No. 2021ZD0301800), the Youth Innovation Promotion Association of CAS (Grant No. Y2021006) and Synergetic Extreme Condition User Facility

(SECUF). J.M.B. was supported by National Research Foundation (NRF) of Korea through Grants No. NRF-2022R1C1C2008671. The work at Brookhaven was supported by the Office of Basic Energy Sciences, U.S. Department of Energy (DOE) under Contract No de-sc0012704.

## Author contributions

X.J.Z., T.X., L.Z., and Q.G. proposed and designed the research. G.D.G., Q.G., and P.A. prepared single crystal. Q.G. carried out the experiment with J.L. and P.A.; Y.Q.C., C.L., Y.W., Q.G., H.T.Y., X.Y.L., C.H.Y., H.C., Z.H.Z., L.Z., G.D.L., F.F.Z., F.Y., S.J.Z., Q.J.P., Z.Y.X., and X.J.Z. contributed to the development and maintenance of Laser ARPES system. Q.G., L.Z., and X.J.Z. analyzed the data. J.M.B., Q.G., and H.Y.C. contributed to model calculations. X.J.Z., L.Z., and Q.G. wrote the paper. All authors discussed the results and commented on the manuscript.

## Competing interests

The authors declare no competing interests.
