## [Peer Review File · Nature Communications]

ARPES Detection of Superconducting Gap Sign in Unconventional SuperconductorsREVIEWER COMMENTS

Reviewer #1 (Remarks to the Author):

The authors report a novel method to turn ARPES, which has been a powerful energy gap size mapping tool in the past few decades, into a phase-sensitive order parameter measurement tool in superconducting cuprates. The key methodological innovation here is the use of Bogoliubov quasiparticle interference effect between intercepting bands from different momentum sectors of opposite order parameter signs. And the crucial technological enabler here is the exceptional energy resolution and spectral quality (probably also in part thanks to the high quality sample/cleave) shown in the data. Despite the fact that such "naturally occurring" interference is arguably rare in other systems, and it appears to take a very fortunate coincidence that Bi-2212 cuprates have the required superstructure, sign-changing order parameter around the superstructure direction, and a sizable energy gap to make the effect detectable, the results are nonetheless an exhilarating demonstration of a completely new way of using ARPES to derive sign-changing order parameters. Barring a few minor comments, I enthusiastically recommend publication of this important work.

- The authors should be very explicit about the limitation of this method: as shown in the present work, it is not possible to determine the full k -dependence of the phase, or pinpoint where the sign-change happens. This method is good for validating or falsifying assumed gap structures, but it will require a continuously tunable band crossing in k -space to truly map out the sign-changing order parameter. Also, I believe there is also a requirement on how far the two bands can be in k -space in order to maintain a visible BQP hybridization - if too far, then the effect will be too weak to be observed.

- the authors should discuss the distinction between the BQP hybridization in the present work (Bi-2212 bands from superstructure folding) and earlier works in trilayer cuprates (bands from inner vs outer CuO₂ layer). In previous discussions, it is demonstrated that such BQP hybridization should not be possible in bilayer compounds between the bonding and antibonding bands of identical gap sizes. Granted that this work discusses a different - but intimately related - situation. To see the observed interference effect, the local superconducting gaps should be different on the two bands according to Eqn(3) in the supplement of Ref[20]; however it is not obvious why this is satisfied in the present study. I'd like to see this point discussed.

- To generalize this method, as the authors discussed in the final section, engineered superstructures are needed. However, one should be more nuanced than just having superstructures. If the superstructure only affects the photoelectron in the post-emission process, then there won't be any initial state hybridization.

- The authors should comment on the lack of particle-hole symmetry in the Fermi function divided data (the statistics is great, but the gap center is massively off of the chemical potential, compared to the gap size itself). The discussion in Fig.S7 of Phys Rev X 11, 031068 (2021) may be relevant? A related question is: if the asymmetry is already this big, how does this affect the extracted gap and its momentum dependence near the node (which is absolutely critical to the main conclusion)?

- How error bars are derived should be stated.

Reviewer #2 (Remarks to the Author):

In this manuscript, Gao et al. proposed a method to probe the sign of the superconducting order parameter using ARPES. The method requires the presence of two bands close enough in energy-momentum space and with strong enough interband coupling. Under such conditions, the authors showed via simulation that the quasiparticle dispersions in the superconducting state will be sensitive to the phase difference between the superconducting order parameters on the two bands.

The authors then validated the method experimentally using the cuprate superconductor Bi2212 as an example. Here, the BiO superstructure causes a periodic potential in the CuO₂ layers that folds the Fermi surface, creating the conditions required by the proposed method near the crossing points of the main and folded Fermi surfaces. Using very-high-resolution ARPES, the authors clearly resolved the dispersions near one of the crossing points and found them to indicate a sign change between the main and folded Fermi surface branches near the crossing point. This reaffirms the d-wave gap symmetry in Bi2212.

I believe this work is important to the broad fields of unconventional superconductivity and ARPES. To determine the pairing symmetry is one of the key tasks in studying unconventional superconductors. Historically, ARPES played a significant but not deterministic role in such studies. Here, the authors demonstrated that ARPES can actually be phase sensitive and deterministic. One might argue that the requirements of the proposed method are hard to be met on a generic material other than Bi2212. However, it should be noted that the key requirement is a perturbative periodic potential that folds the native Fermi surface and thus creates crossing points, which is not far from practical in the age of van-der-Waals heterostructures.

The authors also presented very-high-quality ARPES data, the kind that is only possible from few places in the world. These data not only well support the method the authors proposed, but also deepen our understanding of Bi2212. It was unambiguously shown that the superstructure can create accidental nodes away from the zone diagonal, which may have important implications for understanding recent experiments such as Ref. 25 and 26.

I do have a few comments about technical details as listed below. Once addressed, I believe that the manuscript will be well suited for publication on Nature Communications.

1. The momentum axes of the energy-momentum cuts in Figures 2, 3, and 4 are somewhat confusing. It took me a while to guess that the authors probably used the momentum along the cut direction with the origin defined at the zone corner. It would be good to specify in the captions the definition of the momentum axis.

2. The theoretical simulations need to be described in more detail on the level of Ref. 21. For example, currently the spectral function is defined as a 4x4 matrix, and it is unclear which elements are plotted in the main text figures. It is also unclear what kind of self energy goes into the simulation.

3. For the theoretical simulation specific to Bi2212, currently the main and folded bands are treated as two intrinsic and separate bands. However, the actual situation is that there is only one intrinsic band (or two if one counts bilayer splitting), with its states at k mixed to those at $k+Q$ by the superstructure potential V_Q . The authors' current treatment would give rise to wrong intensity ratios between the main and folded bands, and also make it less clear why the "superstructure band" remembers where its node is instead of simply respecting the nodal line along the zone diagonal. I suggest the authors to incorporate the actual treatment, or at least clarify the limitations of the current treatment to avoid confusions.

4. It is known that in the presence of superstructure, the photoemission intensity at momentum k is the sum of "diffracted intensity" from $k+n*Q$ (which is a final-state effect) and the intrinsic

intensity from the reconstructed initial state. Can the authors comment on whether their analysis can be affected by the diffracted intensity?

Response to Reviewer's Comments

Response to Reviewer #1's Report

The authors report a novel method to turn ARPES, which has been a powerful energy gap size mapping tool in the past few decades, into a phase-sensitive order parameter measurement tool in superconducting cuprates. The key methodological innovation here is the use of Bogoliubov quasiparticle interference effect between intercepting bands from different momentum sectors of opposite order parameter signs. And the crucial technological enabler here is the exceptional energy resolution and spectral quality (probably also in part thanks to the high quality sample/cleave) shown in the data. Despite the fact that such "naturally occurring" interference is arguably rare in other systems, and it appears to take a very fortunate coincidence that Bi-2212 cuprates have the required superstructure, sign-changing order parameter around the superstructure direction, and a sizable energy gap to make the effect detectable, the results are nonetheless an exhilarating demonstration of a completely new way of using ARPES to derive sign-changing order parameters. Barring a few minor comments, I enthusiastically recommend publication of this important work.

We thank the Reviewer for carefully reviewing our paper. The reviewer has nicely captured the main results, high quality of our data, significance and impact of our work. We thank the Reviewer for recommending the publication of our paper. We also appreciate the comments and suggestions from the Reviewer. We have revised the manuscript and discussed these revisions in each of the following comments.

(1). **Reviewer's question:** *"The authors should be very explicit about the limitation of this method: as shown in the present work, it is not possible to determine the full k -dependence of the phase, or pinpoint where the sign-change happens. This method is good for validating or falsifying assumed gap structures, but it will require a continuously tunable band crossing in k -space to truly map out the sign-changing order parameter. Also, I believe there is also a requirement on how far the two bands can be in k -space in order to maintain a visible BQP hybridization - if too far, then the effect will be too weak to be observed."*

Our response: We thank the reviewer for the suggestion. We agree that it is difficult to use our method to determine the full k -dependence of the phase because the utilization of our method requires the two bands are close and have relatively strong interband coupling. As the reviewer said, our method is good for testing the proposed gap structures.

Changes made: Following the suggestion of the reviewer, we added in the revised manuscript more discussions on the limitations of the method: *"Since the effectiveness of the method requires that the two bands are close and have relatively strong interband coupling, it is good for testing the proposed gap structure but is difficult to determine the full momentum-dependence of the phase."*

(2). **Reviewer's question:** *"the authors should discuss the distinction between the BQP hybridization in the present work (Bi-2212 bands from superstructure folding) and earlier works in trilayer cuprates (bands from inner vs outer CuO₂ layer). In previous discussions, it is demonstrated that such BQP hybridization should not be possible in bilayer compounds between the bonding and antibonding bands of identical gap sizes. Granted that this work discusses a different - but intimately related - situation. To see the observed interference effect, the local superconducting gaps should be different on the two bands according to Eqn(3) in the supplement of Ref[20]; however it is not obvious why this is satisfied in the present study. I'd like to see this point discussed."*

Our response: The Bogoliubov band hybridization between the two bands in Bi2212 or between the two bands from the inner and outer CuO₂ layers in Bi2223 can be universally described by the same two-band model in the Eq. (3) in the Supplementary Material of Ref[20]. The extent of the hybridization depends on the separation of the two bands, the interband coupling strength, the superconducting gap size difference on the two bands, and the relative gap sign of the two bands. In Bi2223, the observed strong Bogoliubov band hybridization between the bands from the inner and outer CuO₂ layers can be well understood by the large superconducting gap size difference of the two bands even though the gaps have the same sign [20, 21]. In Bi2212, the main bonding and main antibonding bands have identical gap size[28] and the same gap sign as shown in the first quadrant of the Brillouin zone in Fig. 2a. According to the two-band model, the resulted Bogoliubov band hybridization should be nearly zero. This is consistent with the measured results that no obvious hybridization is observed between the two bands (Fig. S4 in Supplementary Materials). In Bi2212, the main antibonding Fermi surface and the superstructure bonding Fermi surface cross each other in the second quadrant of the Brillouin zone, as shown in Fig. 2a. Near the crossing region, the main antibonding band and the superstructure bonding band have identical gap size but the opposite gap sign. This gives rise to a dramatic Bogoliubov band hybridization between the two bands as shown in Fig. 2 and Fig. 3. It is true that, to see the observed interference effect, the local superconducting gaps should be different on the two bands. In our case, the gap sign is opposite between the main antibonding band and the superstructure bonding band near the crossing region in the second quadrant.

Changes made: Following the reviewer's suggestion, we added the above discussion in the revised manuscript. "The Bogoliubov band hybridization between the two bands in Bi2212 or between the two bands from the inner and outer CuO₂ layers in Bi₂Sr₂Ca₂Cu₃O₁₀ (Bi2223)[20, 21] can be universally described by the same two-band model in Eq. 1. The extent of the hybridization depends on the separation of the two bands, their interband coupling strength, the superconducting gap size difference on the two bands, and their relative gap sign. In Bi2223, the observed strong Bogoliubov band hybridization between the bands from the inner and outer CuO₂ layers can be well understood by the large superconducting gap size difference of the two bands even though the gaps have the same sign[20, 21]. In Bi2212, the main bonding band and the main antibonding band have comparable superconducting gap size[28] and the same gap sign as shown in the first quadrant of the Brillouin zone in Fig. 2a. According to the two-band model, the resulted Bogoliubov band hybridization should be nearly zero. This is consistent with the measured results that no obvious hybridization is observed between the two bands (Fig. S3 in Supplementary Materials). On the other hand, the absence of the Bogoliubov band hybridization between the main bonding band and the main antibonding band indicates that they have the same gap sign. In Bi2212, the main antibonding Fermi surface and the superstructure bonding Fermi surface cross each other in the second quadrant of the Brillouin zone, as shown in Fig. 2a. Near the crossing region, the main antibonding band and the superstructure bonding band have identical gap size but the opposite gap sign. This gives rise to a dramatic Bogoliubov band hybridization between the two bands as shown in Fig. 2 and Fig. 3. In order to see the obvious Bogoliubov band hybridization effect, the local superconducting gaps should be different on the two bands. In the present case, the gap sign is opposite between the main antibonding band and the superstructure bonding band near the crossing region in the second quadrant."

(3). **Reviewer's question:** "To generalize this method, as the authors discussed in the final section, engineered superstructures are needed. However, one should be more nuanced than just having superstructures. If the superstructure only affects the photoelectron in the post-emission process, then there won't be any initial state hybridization."

Our response: We agree with the reviewer that the engineered superstructures should cause the hybridization between the initial bands and the superstructure bands.

Changes made: As the reviewer suggested, we modified the related discussion in the revised manuscript: “In the case that no natural band folding mechanisms are available, it is possible to engineer the band structures to produce hybridization between the initial bands and the superstructure bands such as those realized in twisted graphene [44-46] or substrate-controlled film growth [47, 48].”

(4). **Reviewer’s question:** “*The authors should comment on the lack of particle-hole symmetry in the Fermi function divided data (the statistics is great, but the gap center is massively off of the chemical potential, compared to the gap size itself). The discussion in Fig.S7 of Phys Rev X 11, 031068 (2021) may be relevant? A related question is: if the asymmetry is already this big, how does this affect the extracted gap and its momentum dependence near the node (which is absolutely critical to the main conclusion)?*”

Our response: As pointed out by the reviewer, the particle-hole symmetry is broken in the Fermi function divided data in Fig. 2d as replotted in Fig. R1a. The gap center is obviously away from the chemical potential. This is not caused by the improper Fermi function division as discussed in Phys. Rev. X 11, 031068 (2021) because the energy resolution we used is rather high (~1 meV). Instead, the broken particle-hole symmetry is intrinsic in the case that the gap sign between the two bands is opposite. This can be illustrated more clearly in the simulated data in Fig. 2g as replotted in Fig. R1c where the broken particle-hole symmetry is also obvious. We note that the particle-hole symmetry breaking manifests itself in the dramatic difference of the spectral weight distribution with respect to the chemical potential (Fig. R1d) although the energy positions remain symmetric with respect to the chemical potential.

Fig. R1 **a**, Band structure measured at 15 K in the second quadrant with a Fermi surface angle $\theta=32$ replotted from Fig. 2d in the main text. **b**, EDC at the Fermi momentum as marked by the vertical red line in **a**. **c**, The simulated band structure with a Fermi surface angle $\theta=32$ replotted from Fig. 2g in the main text. **d**, The simulated EDC at the Fermi momentum as marked by the vertical red line in **c**.

The superconducting gap size is determined from the EDC peak position. Since the peak positions remain symmetric with respect to the chemical potential in the case of the particle-hole symmetry

breaking as shown in Fig. R1d and Fig. 4(g,h), it does not affect our determination of the superconducting gap even near the nodal region.

Changes made: Following the reviewer's suggestion, in the revised manuscript, we added more discussions on the particle-hole symmetry breaking and its effect on the superconducting gap determination: "The gap center is obviously away from the chemical potential. This is not caused by the improper Fermi function division[25] because the energy resolution we used is rather high (~1 meV)." and "We note that the particle-hole symmetry breaking manifests itself in the dramatic difference of the spectral weight distribution with respect to the chemical potential but the energy positions remain symmetric with respect to the chemical potential. Since the superconducting gap size is determined from the EDC peak position, the broken particle-hole symmetry does not affect our determination of the superconducting gap even near the nodal region."

(5). **Reviewer's question:** "How error bars are derived should be stated."

Our response: The error bars in extracting the superconducting gap are estimated by considering the uncertainty in determining the peak positions and the experimental energy resolution.

Changes made: In the revised manuscript, we added the following description in the caption of Fig. 4. "The error bars are estimated by considering the uncertainty in determining the peak positions and the experimental energy resolution."

Response to Reviewer #2's Report

In this manuscript, Gao et al. proposed a method to probe the sign of the superconducting order parameter using ARPES. The method requires the presence of two bands close enough in energy-momentum space and with strong enough interband coupling. Under such conditions, the authors showed via simulation that the quasiparticle dispersions in the superconducting state will be sensitive to the phase difference between the superconducting order parameters on the two bands.

The authors then validated the method experimentally using the cuprate superconductor Bi2212 as an example. Here, the BiO superstructure causes a periodic potential in the CuO₂ layers that folds the Fermi surface, creating the conditions required by the proposed method near the crossing points of the main and folded Fermi surfaces. Using very-high-resolution ARPES, the authors clearly resolved the dispersions near one of the crossing points and found them to indicate a sign change between the main and folded Fermi surface branches near the crossing point. This reaffirms the d-wave gap symmetry in Bi2212.

I believe this work is important to the broad fields of unconventional superconductivity and ARPES. To determine the pairing symmetry is one of the key tasks in studying unconventional superconductors. Historically, ARPES played a significant but not deterministic role in such studies. Here, the authors demonstrated that ARPES can actually be phase sensitive and deterministic. One might argue that the requirements of the proposed method are hard to be met on a generic material other than Bi2212. However, it should be noted that the key requirement is a perturbative periodic potential that folds the native Fermi surface and thus creates crossing points, which is not far from practical in the age of van-der-Waals heterostructures.

The authors also presented very-high-quality ARPES data, the kind that is only possible from few places in the world. These data not only well support the method the authors proposed, but also deepen our

understanding of Bi2212. It was unambiguously shown that the superstructure can create accidental nodes away from the zone diagonal, which may have important implications for understanding recent experiments such as Ref. 25 and 26.

I do have a few comments about technical details as listed below. Once addressed, I believe that the manuscript will be well suited for publication on Nature Communications.

We thank the Reviewer for carefully reviewing our paper and providing constructive comments and suggestions to improve our paper. The reviewer has nicely captured the main results, high quality of our data, significance and impact of our work. In particular, we agree with the reviewer that our method can be more general in the age of van der Waals heterostructures. We also thank the Reviewer for recommending the publication of our paper after addressing his/her comments.

Following the reviewer's comments, in the revised manuscript, we added or modified the related discussions. **“It should be noted that the key requirement is a perturbative periodic potential that folds the native Fermi surface and thus creates crossing points. This is not far from practical in the age of van-der-Waals heterostructures[49].”** and **“We have demonstrated that, in addition to measuring the superconducting gap size, ARPES can be phase sensitive and deterministic in pinning down the pairing symmetry that is significant to understand the superconductivity mechanism of unconventional superconductors.”**

(1). **Reviewer's question:** *“The momentum axes of the energy-momentum cuts in Figures 2, 3, and 4 are somewhat confusing. It took me a while to guess that the authors probably used the momentum along the cut direction with the origin defined at the zone corner. It would be good to specify in the captions the definition of the momentum axis.”*

Our response: The momentum axes of the cuts in Figures 2, 3, and 4 are shown with the origin defined at the zone corner.

Changes made: Following the review's suggestion, in the revised manuscript, we have added the following description in the caption of Fig. 2. **“The origin of the momentum axis is defined at the zone corner. We adopt this definition throughout this paper.”**

(2). **Reviewer's question:** *“The theoretical simulations need to be described in more detail on the level of Ref. 21. For example, currently the spectral function is defined as a 4x4 matrix, and it is unclear which elements are plotted in the main text figures. It is also unclear what kind of self energy goes into the simulation.”*

Our response: In our simulations, we used the G_{11} and G_{22} elements of the Green's function $G(\mathbf{k}, \omega)$ to obtain the spectral function. We simulated the spectrum with self-energy of the marginal Fermi liquid to describe the interaction between the electrons.

Changes made: According to the reviewer's suggestion, in the revised manuscript, we added more details of our simulations in the Supplementary Materials. **“The Green's function is obtained from the Hamiltonian:**

$$G(\mathbf{k}, \omega) = (\omega - \Sigma(\mathbf{k}, \omega) - H)^{-1}$$

where $\Sigma(\mathbf{k}, \omega)$ is the electron self-energy.

$$\Sigma(\mathbf{k}, \omega) = \Sigma'(\mathbf{k}, \omega) + i\Sigma''(\mathbf{k}, \omega)$$

We simulate the spectrum with self-energy of the marginal Fermi liquid to describe the interaction between the electrons[4].

$$\Sigma''(k, \omega) = \lambda \sqrt{\omega^2 + (\pi k_B T)^2} + \Gamma_0$$

We ignore the real part of the self energy and use the tight binding model in Eq. 8 to describe the band dispersion. Our simulation well reproduces the electronic structure near the Fermi level. The spectral function is obtained by tracing the imaginary part in the electron channel of the Green's function

$$A(k, \omega) = -\frac{1}{\pi} (Im[G_{11}(k, \omega)] + \kappa \cdot Im[G_{22}(k, \omega)])$$

where $\kappa=0.1$ is the intensity ratio between the superstructure band and the main band.”

(3). **Reviewer’s question:** “*For the theoretical simulation specific to Bi2212, currently the main and folded bands are treated as two intrinsic and separate bands. However, the actual situation is that there is only one intrinsic band (or two if one counts bilayer splitting), with its states at k mixed to those at $k+Q$ by the superstructure potential V_Q . The authors’ current treatment would give rise to wrong intensity ratios between the main and folded bands, and also make it less clear why the “superstructure band” remembers where its node is instead of simply respecting the nodal line along the zone diagonal. I suggest the authors to incorporate the actual treatment, or at least clarify the limitations of the current treatment to avoid confusions.*”

Our response: We thank the reviewer for raising the point. It has been well established experimentally that the superstructure bands in Bi2212 are replicas of the main bands, *i.e.*, except for the intensity difference, the dispersion and the superconducting gap of the superstructure bands all mimic those of the main bands. As illustrated in Fig. R2, in the case of the d -wave superconducting gap, the gap nodes are along the ΓX and ΓY directions. The presence of a superstructure shifts the original band structures by $\pm n \cdot Q$. For example, a new superstructure Fermi surface is formed by shifting the original main Fermi surface with $-Q$ for $n=-1$ as shown in Fig. R2. The superconducting gap remains d -wave for the superstructure Fermi surface, but it is in the shifted Brillouin zone with a new center Γ' . The gap nodes on the superstructure Fermi surface are then along the $\Gamma'X'$ and $\Gamma'Y'$ directions. These are the experimental results that have been well established by the previous ARPES measurements[28].

In our treatment, we took two steps. The first is the formation of the superstructure bands. In this case, the related information is taken from the experimental results, including the intensity ratio between the main band and the superstructure band, the bare band dispersion, and the superconducting gap on the main Fermi surface and superstructure Fermi surface. The second step is to describe the hybridization between the main band and the superstructure band using the standard two band model (Eq. 1 in the main text). We note that this two band model (Eq. 1) does not describe the formation of the superstructure bands. Instead, after the superstructure bands are formed, it describes the band hybridization between the main band and the superstructure band. We agree with the reviewer that our treatment in the second step (Eq. 1) does not produce the intensity ratios between the main and folded bands and the location of the nodes on the superstructure Fermi surface. The intensity ratio between the main band and the superstructure band is taken from the experimental results and put in the Eq. 13 in Supplementary Materials in our simulations. The gap node locations of the main Fermi surface and the superstructure Fermi surface have been considered in the first step, and used in our two band model (Eq. 1).

Fig. R2 Schematic Fermi surface of Bi2212. The main bonding Fermi surface (thick blue and red lines) and the superstructure bonding Fermi surface (thin blue and red lines) are plotted. The superstructure Fermi surface is obtained by shifting the main Fermi surface with a superstructure wave vector \mathbf{Q} along ΓY direction. The blue and red lines represent positive and negative sign of the superconducting gap, respectively.

Changes made: To avoid possible confusions and following the reviewer’s comments, in the revised manuscript, we added the above discussions to the Supplementary Materials: “It has been well established experimentally that the superstructure bands in Bi2212 are replicas of the main bands, i.e., except for the intensity difference, the dispersion and the superconducting gap of the superstructure bands all mimic those of the main bands. As illustrated in Fig. S2, in the case of the d -wave superconducting gap, the gap nodes are along the ΓX and ΓY directions. The presence of a superstructure shifts the original band structures by $\pm n*Q$. For example, a new superstructure Fermi surface is formed by shifting the original main Fermi surface with $-Q$ for $n=-1$ as shown in Fig. S2. The superconducting gap remains d -wave for these superstructure Fermi surface, but it is in the shifted Brillouin zone with a new center Γ' . The gap nodes on the superstructure Fermi surface are then along the $\Gamma'X'$ and $\Gamma'Y'$ directions. These are the experimental results that have been well established by the previous ARPES measurements[1].

In our theoretical simulations, we took two steps. The first is the formation of the superstructure bands. In this case, the related information are taken from the experimental results, including the intensity ratio between the main band and the superstructure band, the bare band dispersion, and the superconducting gap on the main Fermi surface and superstructure Fermi surface. The second step is to describe the hybridization between the main band and the superstructure band using the standard two band model (Eq. 1 in the main text). We note that this two band model (Eq. 1) does not describe the formation of the superstructure bands. Instead, after the superstructure bands are formed, it describes the band hybridization between the main band and the superstructure band.”

(4). **Reviewer's question:** *“It is known that in the presence of superstructure, the photoemission intensity at momentum k is the sum of “diffracted intensity” from $k+n*Q$ (which is a final-state effect) and the intrinsic intensity from the reconstructed initial state. Can the authors comment on whether their analysis can be affected by the diffracted intensity?”*

Our response: It is true that, in the presence of superstructure, the photoemission intensity at momentum k is the sum of “diffracted intensity” from $k+n*Q$ and the intrinsic intensity from the reconstructed initial state. But we think the diffracted intensity has little effect on our analysis. First, the superstructure bands are weak. In our measurements, the observed superstructure band intensity is only a small fraction of the main band for $n=1$ (~10%) and becomes nearly invisible for $n=2$ and larger. Second, in our analysis we are mainly dealing with the electronic states on the Fermi surface or close to the Fermi level. In this case, the electronic states at $\pm Q$ are usually away from the Fermi level, leaving little effect on the electronic states near the Fermi level. As shown in Fig. 3, the observed results along different momentum cuts in Fig. 3b can be well described by the theoretical simulations in Fig. 3e. It also indicates that the diffracted intensity has little effect on our results and analysis.

Summary of Modifications:

1. Following the suggestion of the reviewer, we added in the revised manuscript on page 12 more discussions on the limitations of the method: “Since the effectiveness of the method requires that the two bands are close and have relatively strong interband coupling, it is good for testing the proposed gap structure but is difficult to determine the full momentum-dependence of the phase.”

2. Following the reviewer’s suggestion, we added more discussion in the revised manuscript on page 12. “The Bogoliubov band hybridization between the two bands in Bi2212 or between the two bands from the inner and outer CuO₂ layers in Bi₂Sr₂Ca₂Cu₃O₁₀ (Bi2223)[20, 21] can be universally described by the same two-band model in Eq. 1. The extent of the hybridization depends on the separation of the two bands, their interband coupling strength, the superconducting gap size difference on the two bands, and their relative gap sign. In Bi2223, the observed strong Bogoliubov band hybridization between the bands from the inner and outer CuO₂ layers can be well understood by the large superconducting gap size difference of the two bands even though the gaps have the same sign[20, 21]. In Bi2212, the main bonding band and the main antibonding band have comparable superconducting gap size[28] and the same gap sign as shown in the first quadrant of the Brillouin zone in Fig. 2a. According to the two-band model, the resulted Bogoliubov band hybridization should be nearly zero. This is consistent with the measured results that no obvious hybridization is observed between the two bands (Fig. S3 in Supplementary Materials). On the other hand, the absence of the Bogoliubov band hybridization between the main bonding band and the main antibonding band indicates that they have the same gap sign. In Bi2212, the main antibonding Fermi surface and the superstructure bonding Fermi surface cross each other in the second quadrant of the Brillouin zone, as shown in Fig. 2a. Near the crossing region, the main antibonding band and the superstructure bonding band have identical gap size but the opposite gap sign. This gives rise to a dramatic Bogoliubov band hybridization between the two bands as shown in Fig. 2 and Fig. 3. In order to see the obvious Bogoliubov band hybridization effect, the local superconducting gaps should be different on the two bands. In the present case, the gap sign is opposite between the main antibonding band and the superstructure bonding band near the crossing region in the second quadrant.”

3, On page 13, we modified the related discussion in the revised manuscript: “In the case that no natural band folding mechanisms are available, it is possible to engineer the band structures to produce hybridization between the initial bands and the superstructure bands such as those realized in twisted graphene [42-44] or substrate-controlled film growth [45, 46].”

4, Following the reviewer’s suggestion, in the revised manuscript we added more discussions on the particle-hole symmetry breaking and its effect on the superconducting gap determination. On page 7, we added “The gap center is obviously away from the chemical potential. This is not caused by the improper Fermi function division[25] because the energy resolution we used is rather high (~1 meV).” On page 9, we added “We note that the particle-hole symmetry breaking manifests itself in the dramatic difference of the spectral weight distribution with respect to the chemical potential but the energy positions remain symmetric with respect to the chemical potential. Since the superconducting gap size is determined from the EDC peak position, the broken particle-hole symmetry does not affect our determination of the superconducting gap even near the nodal region.”

5, In the revised manuscript, we added the following description in the caption of Fig. 4 on page 24. “The error bars are estimated by considering the uncertainty in determining the peak positions and the experimental energy resolution.”

6, Following the reviewer’s comments, in the revised manuscript, we added or modified the related discussions on page 13. “It should be noted that the key requirement is a perturbative periodic potential that

folds the native Fermi surface and thus creates crossing points. This is not far from practical in the age of van-der-Waals heterostructures[49].” and “We have demonstrated that, in addition to measuring the superconducting gap size, ARPES can be phase sensitive and deterministic in pinning down the pairing symmetry that is significant to understand the superconductivity mechanism of unconventional superconductors.”

7, Following the review’s suggestion, in the revised manuscript on page 22, we have added the following description in the caption of Fig. 2. “The origin of the momentum axis is defined at the zone corner. We adopt this definition throughout this paper.”

8, According to the reviewer’s suggestion, we added more details of our simulations in the Supplementary Materials on page 5. “The Green's function is obtained from the Hamiltonian:

$$G(\mathbf{k}, \omega) = (\omega - \Sigma(\mathbf{k}, \omega) - H)^{-1}$$

where $\Sigma(\mathbf{k}, \omega)$ is the electron self-energy.

$$\Sigma(\mathbf{k}, \omega) = \Sigma'(\mathbf{k}, \omega) + i\Sigma''(\mathbf{k}, \omega)$$

We simulate the spectrum with self-energy of the marginal Fermi liquid to describe the interaction between the electrons[4].

$$\Sigma''(\mathbf{k}, \omega) = \lambda\sqrt{\omega^2 + (\pi k_B T)^2} + \Gamma_0$$

We ignore the real part of the self energy and use the tight binding model in Eq. 8 to describe the band dispersion. Our simulation well reproduces the electronic structure near the Fermi level. The spectral function is obtained by tracing the imaginary part in the electron channel of the Green's function

$$A(\mathbf{k}, \omega) = -\frac{1}{\pi} (Im[G_{11}(\mathbf{k}, \omega)] + \kappa \cdot Im[G_{22}(\mathbf{k}, \omega)])$$

where $\kappa=0.1$ is the intensity ratio between the superstructure band and the main band.”

9, In the revised Supplementary Materials, we added the more discussions on page 3. “It has been well established experimentally that the superstructure bands in Bi2212 are replicas of the main bands, i.e., except for the intensity difference, the dispersion and the superconducting gap of the superstructure bands all mimic those of the main bands. As illustrated in Fig. S2, in the case of the *d*-wave superconducting gap, the gap nodes are along the ΓX and ΓY directions. The presence of a superstructure shifts the original band structures by $\pm n \cdot Q$. For example, a new superstructure Fermi surface is formed by shifting the original main Fermi surface with $-Q$ for $n=-1$ as shown in Fig. S2. The superconducting gap remains *d*-wave for these superstructure Fermi surface, but it is in the shifted Brillouin zone with a new center Γ' . The gap nodes on the superstructure Fermi surface are then along the $\Gamma'X'$ and $\Gamma'Y'$ directions. These are the experimental results that have been well established by the previous ARPES measurements[1].

In our theoretical simulations, we took two steps. The first is the formation of the superstructure bands. In this case, the related information are taken from the experimental results, including the intensity ratio between the main band and the superstructure band, the bare band dispersion, and the superconducting gap on the main Fermi surface and superstructure Fermi surface. The second step is to describe the hybridization between the main band and the superstructure band using the standard two band model (Eq. 1 in the main text). We note that this two band model (Eq. 1) does not describe the formation of the superstructure bands. Instead, after the superstructure bands are formed, it describes the band hybridization between the main band and the superstructure band.”

10, In the revised manuscript, we added a new figure (Fig. S2) in the Supplementary Materials.

REVIEWERS' COMMENTS

Reviewer #1 (Remarks to the Author):

The authors have allayed the majority of my previous concerns. The explanations are appreciated, although admittedly the proposed text additions may be a bit too verbose to facilitate a smooth reading experience for a general audience. That being said, I have no more fundamental issues. The fact that there IS pronounced BQP hybridization in this Bi-2212 system across the nodal direction is in itself a vivid manifestation of sign-changing order parameter, and I feel this point should be singularly emphasized in the beginning. The explanation to the particle-hole asymmetry is very enlightening, although the technical details may be better placed in the supplement.

One thing I don't feel sufficiently addressed is the description of the how the error bars are derived. The proposed revision, at the current level, remains nebulous and fuzzy that it is impossible for others to fully reproduce the procedure/errorbar even given the raw data.

Reviewer #2 (Remarks to the Author):

I would like to thank the authors for addressing my questions. I do not fully agree with the authors on the last point, but I do believe that their analysis is not affected by diffracted intensity because the analysis relies on peak positions instead of intensity distributions. I support the publication of the manuscript on Nature Communications.

Response to Reviewer's Comments

Response to Reviewer #1's Report

The authors have allayed the majority of my previous concerns. The explanations are appreciated, although admittedly the proposed text additions may be a bit too verbose to facilitate a smooth reading experience for a general audience. That being said, I have no more fundamental issues. The fact that there IS pronounced BQP hybridization in this Bi-2212 system across the nodal direction is in itself a vivid manifestation of sign-changing order parameter, and I feel this point should be singularly emphasized in the beginning. The explanation to the particle-hole asymmetry is very enlightening, although the technical details may be better placed in the supplement.

One thing I don't feel sufficiently addressed is the description of the how the error bars are derived. The proposed revision, at the current level, remains nebulous and fuzzy that it is impossible for others to fully reproduce the procedure/errorbar even given the raw data.

Our response: We thank the Reviewer for carefully reviewing our revised manuscript and recommending the publication of our paper. The reviewer has nicely captured that the BQP hybridization in Bi-2212 system is a vivid manifestation of sign-changing order parameter and the explanation of the particle-hole asymmetry is very enlightening. Following the reviewer's suggestion, we pointed out that we test our proposed method in Bi-2212 system in the revised abstract.

In the revised manuscript, we provided specifications regarding the determination of all error bars.

Response to Reviewer #2's Report

I would like to thank the authors for addressing my questions. I do not fully agree with the authors on the last point, but I do believe that their analysis is not affected by diffracted intensity because the analysis relies on peak positions instead of intensity distributions. I support the publication of the manuscript on Nature Communications.

Our response: We thank the Reviewer for carefully reviewing our revised manuscript and the support for the publication of the manuscript on Nature Communications. The reviewer has nicely captured that our analysis also relies on peak positions. We note that our calculations highly reproduce the intensity distributions in the ARPES spectra.